# Blood Neurofilament Light Chain: The Neurologist’s Troponin?

**DOI:** 10.3390/biomedicines8110523

**Published:** 2020-11-21

**Authors:** Simon Thebault, Ronald A. Booth, Mark S. Freedman

**Affiliations:** 1Department of Medicine and the Ottawa Hospital Research Institute, The University of Ottawa, Ottawa, ON K1H8L6, Canada; 2Department of Pathology and Laboratory Medicine, Eastern Ontario Regional Laboratory Association and Ottawa Hospital Research Institute, University of Ottawa & The Ottawa Hospital, Ottawa, ON K1H8L6, Canada; rbooth@eorla.ca

**Keywords:** neurofilament light chain, biomarkers, multiple sclerosis

## Abstract

Blood neurofilament light chain (NfL) is a marker of neuro-axonal injury showing promising associations with outcomes of interest in several neurological conditions. Although initially discovered and investigated in the cerebrospinal fluid (CSF), the recent development of ultrasensitive digital immunoassay technologies has enabled reliable detection in serum/plasma, obviating the need for invasive lumbar punctures for longitudinal assessment. The most evidence for utility relates to multiple sclerosis (MS) where it serves as an objective measure of both the inflammatory and degenerative pathologies that characterise this disease. In this review, we summarise the physiology and pathophysiology of neurofilaments before focusing on the technological advancements that have enabled reliable quantification of NfL in blood. As the test case for clinical translation, we then highlight important recent developments linking blood NfL levels to outcomes in MS and the next steps to be overcome before this test is adopted on a routine clinical basis.

## 1. Neurofilament Structure and Function

Neurofilaments are neuronal-specific heteropolymers conventionally considered to consist of a triplet of light (NfL), medium (NfM) and heavy (NfH) chains according to their molecular mass [1]. More recent discoveries show that α-Internexin in the central nervous system [2] and peripherin in the peripheral nervous system [3] can also be included in neurofilament heteropolymers. These five proteins co-assemble into the 10 nM intermediate filaments in different combinations and concentrations depending on the type of neuron, location in the axon and stage of development [4].

Each of the neurofilament proteins consists of an amino-terminal domain that is thought to regulate the formation of oligomers [5], a central helical rod domain, and a variable carboxy-terminal domain. The chain-specific C-terminal domains are the main determinants of differences in molecular mass and phosphorylation between subunits. Following synthesis and assembly in the neuron cell body, tetramers of neurofilament proteins are transported bidirectionally along axons by the microtubular apparatus prior to forming a continuously overlapping array that runs parallel to axons. Once formed, in the healthy state, they are remarkably stable for months to years [6].

In mature myelinated axons, neurofilaments are the single most abundant protein [7]. They perform key roles as part of the neuroaxonal scaffold to resist external pressures, determine axonal diameter, indirectly moderate conduction velocity, and act as an attachment for organelles and other proteins [4]. Beyond their primary structural role in axons, mounting evidence indicates that a unique pool of synaptic neurofilament proteins serves dynamic functions beyond static structural support [8]. Changes in neurofilament phosphorylation may be involved in long term potentiation that underpins memory [9] and NMDA receptor stability is dependent on a synaptic scaffold of neurofilament proteins.

## 2. Neurofilament Pathophysiology

Damage to central nervous system (CNS) neurons and physiologic turnover causes neurofilament release. This translates to elevated levels in the cerebrospinal fluid (CSF) and eventually blood, where the concentration reflects the rate of release from neurons (Figure 1, where we focus on NfL). Physiologic degradation of neurofilaments within neurons is proposed to be a combination of ubiquitin-mediated proteasomal and apophagocytotic pathways [10]. Based on the trafficking of other proteins degraded in the CNS, it is likely that partially degraded fragments of neurofilaments drain directly into CSF and blood via multiple routes. These include direct drainage into CSF and blood via arachnoid granulations as well as lymphatic drainage into the subarachnoid spaces and perivascular spaces [11,12]. Several studies have demonstrated strong correlations between blood and CSF NfL, with r values typically ranging from 0.7 to 8 (e.g., [13]). However, our understanding of the kinetics of neurofilament release, distribution and metabolism is incomplete.

Blood–brain barrier permeability itself may be a confounder; neurofilament quotient in blood compared to CSF could be selectively increased following periods of inflammation such as that seen in MS relapse, positively skewing blood NfL levels. Two recent studies on this topic in MS patients present conflicting results [14,15].

Once NfL enters the blood, the half-life is a key consideration with implications on the frequency of disease activity monitoring. In a longitudinal study of NfL levels before and after intrathecal catheter insertion, NfL in both CSF and serum peaked at 1-month post-surgery, returning to baseline after 6 to 9 months [16]. In longitudinally sampled MS patients around the time of relapse, levels increasing 5 months before, peaking at clinical onset, and recovery within 4–5 months [17]. Therefore, quarterly measurement is likely sufficient, a frequency that our group is currently investigating in longitudinal prospective studies of serum NfL.

Age is the principal physiologic covariate of NfL levels. Levels in healthy controls increase by 2.2% per year [18,19]. Furthermore, an inflection point is observable above the age of 60, after which both sNfL levels, as well as the inter-individual variability in levels, increase greatly [20]. It is speculated that these changes are attributable to both aging itself as well as the accumulation of subclinical comorbidities. Other factors outside of neurological disease itself that may alter neurofilament levels include BMI [21] as well as vascular risk factors [22].

Although the primary focus of this review is the pathophysiologic relevance of NfL concentrations as they relate to neurological diseases such as MS, the vital role of neurofilaments is underlined by various human mutations that interfere with their function and homeostasis. Mutations of gigaxonin, a key component in the ubiquitin-dependent intermediate filament degradation, results in the pathological aggregation of neurofilaments in neurons and a severe neurodegenerative condition called giant axonal neuropathy [23]. Mutations in the neurofilament light chain gene itself result in axonal forms of hereditary motor sensory polyneuropathy [24] and variants of the heavy neurofilament subunit are associated with the development of amyotrophic lateral sclerosis [25].

Intriguingly there is evidence that autoimmunity can be directed against neurofilament proteins themselves [26,27,28,29,30]. CSF from MS patients contain anti-NfL antibodies [26] and these antibodies co-localise with neurons in human MS pathological lesions [27]. The pathogenic potential of anti-NfL antibodies is a topic of debate as the neurofilament light chain is intracellular and presumably not amenable to immune surveillance or targeting in the healthy state. However, intrathecally transferred anti-NfL antibodies in rodent models of experimental autoimmune encephalomyelitis (EAE) results in disability progression [28]. Anti-NfL antibody concentration also appears to correlate with MRI tissue damage, particularly lower brain volumes [29]. Following effective treatment of MS with natalizumab, anti-NfL antibody concentrations decrease [30]. Although the pathogenic potential of antibodies directed against an intracellular antigen such as NfL remains debatable, these circulating antibodies could also have important and unexplored implications on neurofilament metabolism in the periphery as well as interference in NfL assays which are antibody-based immunoassays.

## 3. Measurement of Blood Levels of Neurofilament Light Chain

Of the family of neurofilament proteins, neurofilament light chain (NfL) has gained the most interest as a candidate marker of outcomes in neurological diseases. This was not without contention. While the neurofilament light chain is the most abundant and soluble of the neurofilament proteins, phosphorylated neurofilament heavy chain (NfH) was initially thought to be more resistant to protease activity [31,32,33]. NfL was thought to be unstable in vitro [34] and initial research focused on NfH quantified by enzyme-linked immunosorbent assay (ELISA) or electrochemiluminescence (ECL) as a biomarker of axonal damage in MS [35,36]. However, in 2013, a comparative study of NfL and NfH found both proteins showed equivalent stability after several days at room temperature and through freeze–thaw cycles [37]. Moreover, although some of the differences observed may correspond to analytical methodologies, this study found that NfL levels were higher than NfH and NfL was a better discriminator of MS patients from controls.

Initial studies looking at NfL in association with neurological disease outcomes focused on CSF measurements. Although CSF is “closer” to the CNS pathologies (e.g., MS) and NfL concentration is approximately 500-fold higher, the inconvenient and invasive lumbar puncture required severely limits its clinical utility as a frequent serial biomarker. Concentrations in the blood however were too low to be reliably measured with conventional immunoassays such as ELISA or ECL assays. It was not until recently, with the development of the Single-Molecule Assay (SiMoA), that analytical methods become sufficiently sensitive to measure the single-digit picogram per milliliter levels present in blood [38]. This SiMoA technology, similar to other immunoassays, is based on fluorescent microbeads coated with high-affinity capture antibodies that bind NfL followed secondly by a fluorescently labelled detector antibody [39]. The increased analytical sensitivity of the SiMoA assay is due to its unique method of detection. Assay beads with captured NfL and fluorescent detector antibody, are loaded onto an assay disk containing >200,000 microwells capable of holding only a single bead. At high analyte concentration, the total fluorescence can be captured in the traditional manner (analog) and correlated to the analyte concentration. At low analyte concentrations, rather than detecting total fluorescence, a digital image is captured that enumerates individual fluorescent microwells in a binary fashion, effectively lowering the limit of quantitation to the femtomolar range. Although there are several neurofilament assays in development based on other technologies, including widely used chemiluminescent-based assay [40], the data we present here were exclusively generated using the SiMoA platform.

## 4. Blood NfL in Neurological Diseases

As a neuron-specific marker of neurological injury, elevated NfL levels can be found in a variety of conditions that involve neuroaxonal injury in both the central and peripheral nervous system (reviewed by Khalil et al., 2018 [41]). In purely neurodegenerative diseases, NfL could serve as both a prognostic marker of decline but also an efficacy biomarker of experimental therapies. In a meta-analysis of Alzheimer’s disease, frontotemporal and amyotrophic lateral sclerosis [42], plasma NfL levels were elevated in patients compared to controls with utility in differentiating neurodegenerative conditions from non-neurodegenerative mimics. However, due to a lack of specificity to any particular flavor of neurodegeneration, its role as a diagnostic marker is limited. The exception to this is in amyotrophic lateral sclerosis where the uniquely rapid neurodegeneration that characterises this condition results in blood NfL levels several times higher than both controls and other forms of neurodegeneration, and hence may play are role in diagnosis.

In more indolent neurodegenerative processes, early prognostication is an important clinical role. In a study of carriers and non-carriers of autosomal dominant Alzheimer’s disease, the trajectory of NfL in affected individuals compared to controls became segregated during their 30’s, long before clinical onset [43]. Conversely, in Parkinson’s, a condition characterised by particularly slow neurodegeneration, differences between controls and patients are especially small, and the separation NfL trajectories only become apparent after the age of 70 [44] at which point marked disability is usually apparent. Meanwhile, for more acute neuronal injury, neurofilament may also have utility in stroke and traumatic brain injury prognostication. Following a stroke, blood NfL takes several hours to rise, limiting their utility in the hyperacute setting, however, the presence of elevated NfL may have utility in diagnosing subacute strokes as well as in the prognostication of outcomes [45,46]. Similarly, following traumatic brain injury, the extent of NfL increase acutely is predictive of the severity of injury, and while NfL decreases over time, it remains elevated relative to controls several years after the injury. Of recent topical interest, NfL was found to be subtly elevated in the serum of mild-moderate COVID-19 patients [47]. While some have used this evidence to bolster theories of direct neuronal invasion by the virus, these subtle differences could also be attributed to cerebral hypoxia induced by the respiratory virus.

## 5. Blood NfL in MS

Multiple sclerosis is the most common neurological autoimmune disease, known for its varied clinical presentations and unpredictable clinical course [48]. Over the last few decades, there has been a dramatic expansion in the number of immunosuppressive therapies on offer to prevent damaging bouts of focal inflammation and demyelination that characterise this condition. However, a victim of its own success, objective disease monitoring biomarkers are lacking, and adjuncts that can help neurologists track and personalise treatments are sorely needed. Regular MRI scanning remains the gold standard means of detecting sub-clinical inflammatory lesions [49,50], but this costly and inconvenient test has a number of shortcomings. It is poorly predictive of future activity (Ontaneda and Fox, 2017), lacks sensitivity [51], tends to be focused mainly on the brain (leaving out the spinal cord and optic nerves) and entails a large degree of technical variation and subjective interpretation. Notable examples of fluid biomarkers that are already in clinical use in MS include oligoclonal bands [52] (now part of MS diagnostic criteria), antibodies against aquaporin-4 [53] and myelin oligodendrocyte protein [54] (which define pathologically discrete disease entities which previously fell under the umbrella of MS), as well as serological assays for JC virus [55] (pre-immunosuppression risk stratification). However, as of yet, no fluid biomarker has established clinical use in routine disease monitoring and prediction. As a result of this unmet need, associations of blood neurofilament have been intensely studied in MS. In the last two years, more than 200 studies have contributed to a groundswell of evidence associating NfL with outcomes related to disease activity, progression, treatment response and prognosis.

As a cross-sectional measure in groups of MS patients, the strongest evidence links high NfL levels with inflammatory endpoints such as relapses and MRI lesions [19,56,57,58,59,60,61]. This is perhaps counter-intuitive, as one might expect this neuroaxonal protein to associate most strongly with outcomes to neurodegeneration and disease progression. However, axonal damage and loss are hallmarks of demyelinating MS lesions even early-on [62], presumably reflected in a transient marked elevation of NfL.

Other than inflammatory disease activity, an interrelated facet of MS pathology is neurodegeneration and progressive disability accrual. Conveniently, this facet of MS pathology is also objectified with NfL measurement. Patients with progressive MS have higher levels than age and sex matched patients with relapsing disease [63]. Associations can be found between high blood NfL levels and poorer disease progression outcomes including disability scores, conversion to a secondary progressive phenotype, MRI atrophy, and measures of cognitive function [13,18,57,61,64,65,66,67,68,69].

Several groups have also studied the longitudinal significance of blood NfL as a serial disease monitoring/treatment response measure in prospective cohorts from clinical trials. Studies exploring the relationship of NfL kinetics with clinical relapse showed elevations beginning approximately 5 months prior to relapse with a peak at clinical onset and recovery within 4–5 months of remission [17]. When profiled longitudinally, “peaks” of NfL (more than three standard deviations above steady-state) were associated with nearly 80% of MRI and clinical disease activity.

In response to treatment, often related to the availability of retrospective sample sets from well-characterised groups of patients involved in seminal studies, longitudinal NfL reductions have been reported for most established treatments for relapsing MS. These include injectable therapies [18,56,59,70], dimethyl fumarate [71], fingolimod [61,70,72], natalizumab [73], rituximab [74], ocrelizumab [75], ofatumumab [76], alemtuzumab [17] and hematopoietic stem cell transplantation [13]. Encouragingly, reductions in NfL seen after different treatments broadly fall in line with the perceived hierarchy of treatment efficacies, with the greatest reductions seen following the most intensive treatments. Accordingly, in a recent Swedish cohort study of more than 1000 patients receiving one of 6 treatments, the largest reductions in plasma levels were seen following alemtuzumab (48%), and the smallest reduction for teriflunomide (7%), with the other agents falling in the middle [56].

For secondary progressive MS, reductions in NfL have been shown following siponimod [77], ocrelizumab [78] and natalizumab [79]. In primary progressive disease, reductions were seen following fingolimod [80] and ocrelizumab [78]. Given the lack of useful biomarkers otherwise for progressive types of MS, NfL is increasingly seen as an important secondary endpoint in phase 2 and 3 studies of treatments [81].

Many groups have shown the value of NfL in the prediction of future relapses, MRI disease activity, disability worsening, MRI brain and spine atrophy and poorer cognitive outcomes [18,19,59,60,82,83,84]. In a 5-year longitudinal study of more than 1200 Swiss MS patients, high age-adjusted NfL was associated with increased risk of relapse and new MRI activity in the following year [63]. Even in patients who met criteria for “no evidence of disease activity” [85], higher NfL was independently associated with increased risk of clinical and/or MRI disease activity in the next year, indicating that NfL is capable of predicting subclinical disease activity otherwise not captured. NfL may also have utility in long term prediction, with 2 separate studies finding an association of early NfL measurements with clinical and MRI disease outcomes more than a decade later [86,87].

While NfL is the closest blood-based disease monitoring marker to clinical translation in MS, there are also other promising candidates that may provide additional information. For instance, glial fibrillary acidic protein (GFAP) is a marker of astrocytic turnover or damage that may serve more as a marker of disease progression [88]. Although not a fluid biomarker, ocular coherence tomography (OCT) peripapillary retinal nerve fibre layer thickness also seems to be useful as a biomarker for the prediction of disability progression [89]. Thus, NfL may represent the first of several fluid biomarkers with relevance in MS monitoring and prediction; one-day multimodal composite indices could be used to most accurately objectify different components of an individual patient’s disease and inform treatment decisions.

## 6. Conclusions

Measurement of a convenient objective blood marker of neuronal injury in patients is an appealing prospect for neurologists. Analogous to the cardiologist’s troponin, neurofilament light chain is a structural axonal protein that can be detected in the blood at elevated levels in a variety of neurological disease states which can be followed longitudinally. Enabled by recent advancements in assay technologies, many consider this test to be on the verge of clinical translation in a number of different settings. Given its neuron-specific nature, but lack of disease specificity, on its own it is not a helpful diagnostic marker. However, in defined neurological conditions that require monitoring, in particular MS where we have treatments to offer, NfL is rapidly gaining traction.

It seems likely that MS will represent the test case for the clinical translation of blood NfL, where it will be a greatly-needed adjunct to clinical and MRI assessment. While we already know that elevated NfL is concerning and low NfL is reassuring, a number of challenges remain before this test is ready for widespread adoption. Foremost amongst them is the need for age-adjusted normative datasets and cutoff values so that physicians can better interpret individual patient results. Key elements of neurofilament kinetics in the blood, such as blood half-life, need to be delineated to inform optimal testing frequency in clinical practice. Additionally, the ongoing efforts of multisite validation efforts will be important in standardising measurement between clinical laboratories and ensuring that any concerns of analytical validity are allayed. Nonetheless, many are optimistic that NfL could represent the first of its kind in neurology: a broadly-applicable protein biomarker that objectively reflects underlying pathology which can be harnessed to improve patient outcomes.

## Figures and Tables

**Figure 1 biomedicines-08-00523-f001:**
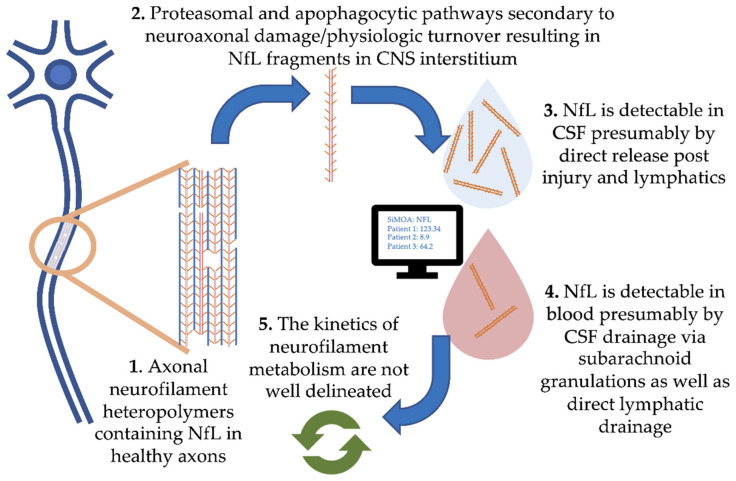
Pathophysiology of neurofilament light chain in blood and cerebrospinal fluid (CSF).

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
