# Peer review of "Blood Neurofilament Light Chain: The Neurologist’s Troponin?"

_biomedicines, 2020, doi:10.3390/biomedicines8110523_

Round 1
Reviewer 1 Report
In this paper the authors explain why the light chain neurofilament (NfL) dosage in the blood can be useful in multiple sclerosis.
We would welcome references to the diagnostic markers currently in use, even if they are unsatisfactory, and to possible studies in experimental diseases. What other markers could the NfL dosage be associated with? Is there information on any correlation with inflammatory markers?
Other comments in detail:
1) The title should be changed to indicate that the study is about Multiple Sclerosis;
2) figure 1 needs to be adjusted;
3) line 71, after themselves, put a reference.
4) line 71: clarify what "these antibodies" refers to
5) line 102: write fluorescent, instead of florescent.
Author Response
Many thanks: Please see our response attached

Reviewer 2 Report
The manuscript is a review that summarizes the physiology and pathophysiology of neurofilaments, the technic of measurement of blood levels of neurofilament light chain and clinical relevance. The topic of the manuscript is important, regarding the continuous development of these measurements in clinical trials. The manuscript is well written.
I only have a couple of minor comments and suggestions:
Figure 1 requires slight modification, because parts of it are not visible.
The key element of this manuscript is the discussion of NFL, as it becomes a potential biomarker to predict disease progression and response to treatment in MS. What is the suggested frequency of monitoring of these levels to evaluate the efficacy of DMT in MS?
The authors would increase the value/utility of this paper if they added extra information to data of the blood levels of NFL, (duration of treatment, change in NfL concentration), including in a Table, because some of papers cited (69-72) can not be found in PubMed.
Author Response
Many thanks for your comments
Please find our responses attached
